# Artificial Generational Intelligence: Cultural Accumulation in Reinforcement Learning

**Jonathan Cook** *
FLAIR, University of Oxford
jonathan.cook2@hertford.ox.ac.uk

**Chris Lu** *
FLAIR, University of Oxford
christopher.lu@eng.ox.ac.uk

**Edward Hughes**
Google DeepMind
edwardhughes@google.com

**Joel Z. Leibo**
Google DeepMind
jzl@google.com

**Jakob Foerster**
FLAIR, University of Oxford
jakob@eng.ox.ac.uk

## Abstract

*Cultural accumulation* drives the open-ended and diverse progress in capabilities spanning human history. It builds an expanding body of knowledge and skills by combining individual exploration with inter-generational information transmission. Despite its widespread success among humans, the capacity for artificial learning agents to accumulate culture remains under-explored. In particular, approaches to reinforcement learning typically strive for improvements over only a *single* lifetime. Generational algorithms that do exist fail to capture the open-ended, emergent nature of cultural accumulation, which allows individuals to trade-off innovation and imitation. Building on the previously demonstrated ability for reinforcement learning agents to perform social learning, we find that training setups which balance this with independent learning give rise to cultural accumulation. These accumulating agents outperform those trained for a single lifetime with the same cumulative experience. We explore this accumulation by constructing two models under two distinct notions of a generation: *episodic generations*, in which accumulation occurs via *in-context learning* and *train-time generations*, in which accumulation occurs via *in-weights* learning. In-context and in-weights cultural accumulation can be interpreted as analogous to knowledge and skill accumulation, respectively. To the best of our knowledge, this work is the first to present general models that achieve emergent cultural accumulation in reinforcement learning, opening up new avenues towards more open-ended learning systems, as well as presenting new opportunities for modelling human culture.

## 1 Introduction

The capacity to learn skills and accumulate knowledge over timescales that far outstrip a single lifetime (i.e., across generations) is commonly referred to as *cultural accumulation* [Hofstede et al., 1994, Tennie et al., 2009]. Cultural accumulation has been considered the key to human success [Henrich, 2019], continuously aggregating new skills, knowledge and technology. It also sustains generational improvements in the behaviour of other species, such as in the homing efficiency of birds [Sasaki and Biro, 2017]. The two core mechanisms underpinning cultural accumulation are *social learning* [Bandura and Walters, 1977, Hoppitt and Laland, 2013] and *independent discovery* [Mesoudi, 2011]. Its effectiveness can be attributed to the flexibility with which participants engage in these two mechanisms [Enquist et al., 2008, Rendell et al., 2010, Mesoudi and Thornton, 2018].

---

*Equal contribution.

38th Conference on Neural Information Processing Systems (NeurIPS 2024).

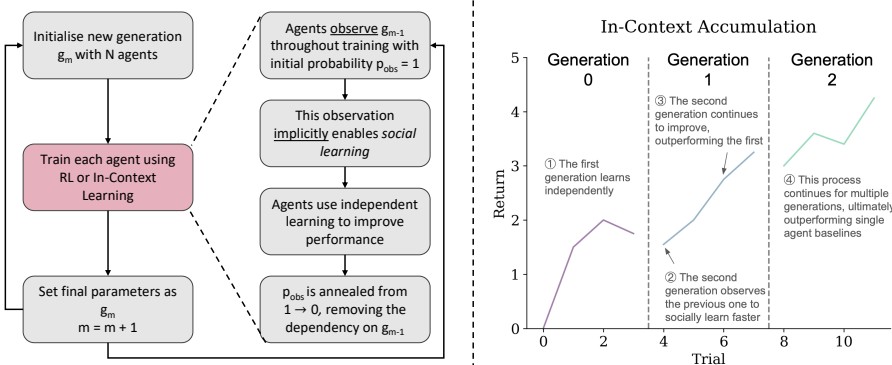

Figure 1: **Left**: A flow chart describing our RL model of cultural accumulation. **Right**: An annotated, illustrative plot demonstrating in-context accumulation as observed in our results.

Cumulative culture thus executes an evolutionary search in the space of behaviours, with higher rates of independent discovery corresponding to a higher mutation rate. Ultimately, how individuals should balance social vs. independent learning depends on the validity and fidelity of social learning signals, i.e., how successful other agents in the environment are at the same task.

Given the profound success of cultural accumulation in nature, it is natural to explore its applicability to artificial learning systems, which remains an under-explored research direction. In deep reinforcement learning (RL), the learning problem is usually framed as taking place over a single "lifetime". Generational methods that do exist, such as iterated policy distillation [Schmitt et al., 2018, Stooke et al., 2022] and expert iteration [Anthony et al., 2017], explicitly optimise new generations with hand-crafted and explicit imitation learning techniques, as opposed to learning accumulation implicitly, which should ultimately enable greater flexibility. Prior work has successfully demonstrated emergent social learning in RL [Borsa et al., 2017, Ndousse et al., 2021, Bhoopchand et al., 2023], showing that it helps agents solve hard exploration problems and adapt online to new tasks. However, these works have only considered one iteration of information transmission from an expert to a student. In this work, we therefore investigate how social learning and exploration can be balanced to achieve cultural accumulation in RL. This would lay the foundations for an open-ended, population-based self-improvement loop among artificial learning agents. It would also establish new tools for modelling cultural accumulation.

In humans, cultural accumulation takes place over a number of timescales due to the different rates at which knowledge, skills and technology are acquired [Perreault, 2012, Cochrane et al., 2023]. We therefore present two corresponding formulations of cultural accumulation in RL: *in-context* accumulation, which operates over fast adaptation to new environments, and *in-weights* accumulation, which operates over the slower process of updating weights (i.e., training). The in-context setting can be interpreted as analogous to short-term knowledge accumulation and the in-weights setting as analogous to long-term, skills-based accumulation.

We demonstrate the effectiveness of both the in-context and in-weights models by showing sustained generational performance gains on several tasks requiring exploration under partial observability[2]. On each task, we find that accumulating agents outperform those that learn for a single lifetime of the same total experience budget. This cultural accumulation emerges purely from individual agents maximising their independent rewards, without any additional losses.

## 2 Background

### 2.1 Partially-Observable Stochastic Games

A Partially-Observable Stochastic Game (POSG) [Kuhn, 1953] is defined by the tuple $M = \langle I, S, A, T, R, O, \gamma \rangle$. $I = \{1, \ldots, N\}$ defines the set of agents, $S$ defines the set of states, and $A$ defines the direct product of action spaces for each agent, i.e., $A = A^1 \times \cdots \times A^N$ where $A^i$ is the action space for agent $i \in I$. $T$ defines the stochastic transition function, $T : S \times A \times S \to [0, 1]$, $R$ de-

---

[2]Code can be found at `https://github.com/FLAIROx/cultural-accumulation`.

fines the reward function, $R : S \times A \to \mathcal{R}$, and $O$ defines the observation function, $O : \mathcal{O} \times S \to [0, 1]$ where $\mathcal{O}$ is the observation space. $\gamma \in (0, 1]$ refers to the discount factor of returns.

At each timestep $t$, each agent $i$ samples an action from its policy $a_t^i \sim \pi_{\theta^i}(o_t^i)$, where $\theta_t^i$ corresponds to the policy parameters for agent $i$. These actions are aggregated to form the joint action $a_t = (a_t^1, \ldots, a_t^N)$. The environment then calculates the next state $s_{t+1} \sim T(s_t, a_t)$. The agents then receive the next observation $o_{t+1} \sim O(s_{t+1})$ and reward $r_t \sim R(s_{t+1})$ from the environment. The objective for each agent $i$ in a POSG is to maximise its expected discounted sum of returns $J(\pi_{\theta^i}) = \mathbb{E}_{s_0 \sim \tilde{S}, a_{0:\infty} \sim \pi(s_{1:\infty}^i) \sim T} \left[ \sum_{t=0}^{\infty} \gamma^t R(s_t, a_t) \right]$.

## 2.2 Partially-Observable Markov Decision Processes

A Partially-Observable Markov Decision Processes (POMDP) [Kaelbling et al., 1998] is a special case of the POSG with only one agent. For a given agent $i$ in a POSG, if the policy parameters of all other agents $\theta^{-i}$ are fixed, the POSG can be reduced to a POMDP for agent $i$. Informally, this can be done by assuming the other agents are part of the environment. In other words, sampling from other agent policies would be part of the transition function.

To solve POMDPs, we make use of memory-based policy architectures that represent the prior episode interactions at timestep $t$ using $\phi_t$. Memory is needed to solve POMDPs because of the state aliasing introduced by partial-observability. For recurrent neural networks (RNNs) $\phi_t$ refers to the hidden state at timestep $t$. We now condition the policy on this state $a_t^i \sim \pi_{\theta^i}(o^i | \phi_t^i)$ and also update the hidden state with recurrence function $h_{\theta^i}$, such that $\phi_{t+1} = h_{\theta^i}(o^i | \phi_t^i)$.

## 2.3 Meta-RL

In this paper, we consider meta-RL settings where an agent is tasked with optimising its return over a distribution of POMDPs $\mathcal{M}$. At each episode[3], a POMDP is sampled $M \sim \mathcal{M}$ and the agent is given $K$ trials to optimise its cumulative return. This setting has also been called in-context RL [Laskin et al., 2022]. We adopt the setting of Ni et al. [2021], where meta-RL is formulated as another POMDP $\mathcal{M}$, in which the selection of $M$ is now a part of the state $s$. The agent's last action $a_{t-1}$, reward $r_{t-1}$ and trial termination are appended to the observation $o_t$.

One common approach to meta-RL uses gradient-based optimisation to train a meta-policy with respect to $\mathcal{M}$ using recurrent network architectures [Duan et al., 2016, Wang et al., 2016]. Within each episode, updates to the agent's internal state $\phi$ enable static agent parameters to implement an in-context RL procedure, allowing the agent to adapt to a specific $M$. During adaptation to a new POMDP, this procedure aggregates and stores information from sequential observations. Over long enough episodes, agents can therefore face an in-context exploration-exploitation tradeoff, where within-episode exploration of the environment could yield higher returns.

## 2.4 Generational Training

Generational training refers to training populations of agents in sequence, thus chaining together multiple learning processes [Stooke et al., 2022]. We define a generation $g_m = \{g_m^1, \ldots, g_m^{N_{\text{pop}}}\}$ as a population of $N_{\text{pop}}$ agents. Each $g_m^n$ can therefore be model weights, or an internal state in the case of in-context RL. We distinguish $N_{\text{pop}}$ from the $N$ used in Section 2.1, because each member of a generation of size $N_{\text{pop}}$ may be within its own POSG of $N$ agents. We use $n$ to index a specific member of a generation. A new generation is generated by a process that conditions on the old generation $g_{m+1} = f(g_m)$. By modelling cultural accumulation over both in-context RL and RL training, $f$ could capture generational accumulation in both knowledge and skills.

# 3 Problem Statement

In this work, we investigate how to achieve cultural accumulation in RL agents. We formalise a measure of success in Equation 1, where we compare the returns achieved in a held-out set of environments when learning is spread across $G$ total generations to learning in a single lifetime with

---

[3] Our use of "episode" and "trial" aligns with Bauer et al. [2023], which is the reverse of Duan et al. [2016].

the same total experience budget. This comparison evaluates whether our models capture the benefits of cultural accumulation over singe lifetime learning.

$$R_T(\pi_G, \mathcal{M}|f(\pi_{G-1}|\dots,\pi_1)) > R_{G\cdot T}(\pi_1, \mathcal{M}) \tag{1}$$

Policies are parameterised according to $\pi_\theta(\cdot|\phi)$ where $\theta$ are parameters of the trained network and $\phi$ is the agent's internal state. We define **in-context accumulation** as cultural accumulation during online adaptation to new environments. In this setting, $\theta$ are frozen and $T$ is the length of an episode. We assume that a meta-RL training process has produced $\theta$ and that the internal state $\phi$ is used to distinguish between generations, which we elaborate on in Section 4.1.2. We define **in-weights accumulation** as cultural accumulation over training runs. Here, $T$ is the number of environment steps used for training each generation and each successive generation is trained from randomly initialised parameters $\theta$, meaning that $\theta$ instead are the substrate for accumulation.

### 3.1 Environments

**Memory Sequence:** To study processes of cultural accumulation among humans, Cornish et al. [2017] use a random sequence memorisation task in which sequences recalled by one participant become training data for the next in an iterated learning process. In doing so, they show a cumulative increase in recalled string length, thus simulating cultural accumulation. We thus present *Memory Sequence* as a simple environment for investigating basic cultural accumulation in RL agents. In *Memory Sequence*, there is an arbitrary sequence made up of digits that agents must infer over the course of in-context learning or training. Each action corresponds to predicting a digit. Episodes are fixed length and agents receive $+1$ reward for predicting the correct next digit and $-1$ reward for predicting an incorrect next digit. The sequence is randomly sampled and fixed during in-context adaptation for in-context accumulation, whilst it is fixed across training for in-weights accumulation.

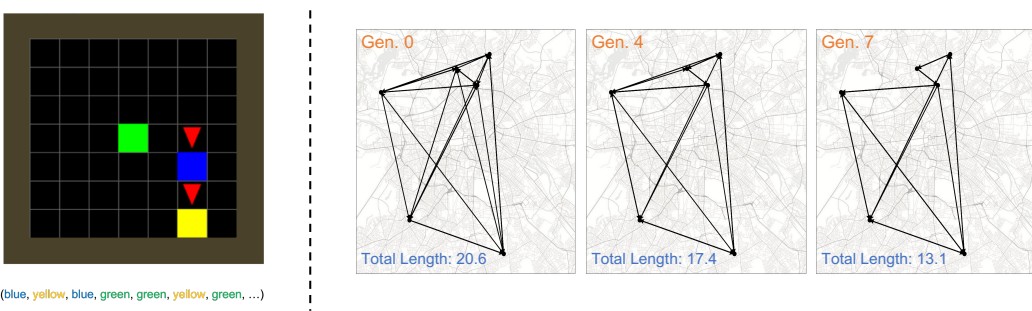

Figure 2: **Left**: A visualization of the Goal Sequence Environment. **Right**: Routes travelled get shorter across generations in the TSP environment. Visualisation implementation is based on Jumanji [Bonnet et al., 2024].

**Goal Sequence:** Previous work on social learning in RL uses a gridworld *Goal Cycle* environment [Ndousse et al., 2021]. In this environment, there are $n$ identical goals and agents must figure out the correct cyclic ordering in which to navigate through them. Agents receive an egocentric partial observation of the environment state on each timestep. Other prior work uses a 3-dimensional *Goal Cycle* environment [Bhoopchand et al., 2023]. *Goal Cycle* is limited in its ability to evaluate generational improvement across learned agent behaviours, because once an agent has identified the correct cyclic order, it should simply repeat the same action sequence of cycling through the goals for the duration of an episode. We therefore introduce *Goal Sequence* (Figure 2) as a simple adaptation of *Goal Cycle*, but with more open-ended properties. Specifically, we replace the cycle with a non-repeating sequence of goals that agents must discover by navigating based on egocentric, partial observations. We use a $7 \times 7$ grid with 3 goal types and agents observe a $3 \times 3$ grid of cells directly in front of them. For in-context accumulation, the goal sequence and positions are randomly generated and fixed across in-context adaptation. For in-weights accumulation, the goal sequence is fixed across training, but goal positions are still randomised between episodes.

**Travelling Salesperson:** Finally, we use the Travelling Salesperson Problem (TSP). Given that cultural accumulation involves the transmission of information not otherwise immediately accessible

to new generations, we specifically focus on the partially-observable variant of TSP [Buck and Keller, 2008, Noormohammadi-Asl and Taghirad, 2019] in which some or all of the cities' coordinates are not observed. The agent's objective is to visit every city, minimising the total length of the route travelled. The positions of cities are randomly generated and fixed across in-context adaptation for in-context accumulation. City positions are fixed across all of training for in-weights accumulation.

## 4   Cultural Accumulation in RL

We model cultural accumulation as taking place within POSGs. Each member of a given generation $g_{m+1}$ learns in parallel on an independent instance of the environment, which also contains *static* members of the previous generation, $g_m$. Therefore, from the perspective of a given agent $n$ in $g_{m+1}$ this is a POMDP, since the prior, static generation can be considered part of the environment. In this work, we focus on *generational improvement* and have thus introduced a separation between the *development phase* of an agent, where they are learning from and improving upon prior generations, and the *transmission phase*, where they are simply acting and being observed by the next generation. Under this formulation, we propose and investigate two distinct mechanisms for cultural accumulation in RL agents based on our *in-context* and *in-weights* dichotomy introduced in Section 3.

### 4.1   In-Context Accumulation

We first consider in-context accumulation, where we aim to achieve cultural accumulation via fast in-context adaptation following a separate phase of meta-RL training. Meta-RL training can be viewed as imbuing agents with multiple memory systems [Squire et al., 1993]. By updating $\phi$, in-context RL executes a processes of knowledge acquisition, or *declarative* memory [Squire, 2004] updating. In-context accumulation extends this knowledge acquisition to operate across multiple generations. The corresponding results are in Section 5.1.

#### 4.1.1   Training

---

**Algorithm 1** Training Loop for In-Context Accumulation (changes to RL$^2$ in red)

---

1:  $\hat{\phi} :=$ init. agent state
2:  $\hat{\theta} :=$ init. agent parameters
3:  $\theta^0 :=$ oracle parameters
4:  $\epsilon :=$ oracle noise factor
5:  $\theta^n \leftarrow \hat{\theta}$ // Initialise the agent parameters
6:  **for** train step $t \in [0, T-1]$ **do**
7:      $s \sim \hat{S}$
8:      $\phi^n \leftarrow \hat{\phi}$
9:      $B :=$ trajectory buffer
10:     **for** each trial $k \in [0, K-1]$ **do**
11:         $p_{\text{obs}} = 1 - k/(K-1)$
12:         **while** trial not done **do**
13:             $o^n \sim O(s)$
14:             **if** $\texttt{IsVisible} \sim \text{Bernoulli}(p_{\text{obs}})$ **then**
15:                 Set oracle visible in $o^n$
16:             **end if**
17:             $a^0 \sim \pi_{\theta^0, \epsilon}(s), a^n \sim \pi_{\theta^n}(o^n|\phi^n)$
18:             $\phi^n \leftarrow f_{\theta^n}(o^n|\phi^n)$
19:             $s, r \sim T(s, a)$
20:             $B \leftarrow (o^n, a^n, r^n)$ // Append transition to buffer
21:         **end while**
22:     **end for**
23:     $\theta^n \leftarrow \text{update}(\theta^n, B)$
24: **end for**

---

To train agents capable of in-context accumulation, we require three attributes: **(1)** agents must learn to learn from the behaviour of other agents (i.e., social learning) within the context of a single episode,

**(2)** agents must be able to act independently by their last trial, so that their behaviour can provide useful demonstrations for the next generation, **(3)** agents must use sufficient independent exploration to improve on the the previous generation, or each new generation will only be as good as the last.

To facilitate **(1)** during training, but *not* at test time, we assume access to an oracle $O$ with fixed parameters $\hat{\theta}^0$. $O$ is able to condition on the full state $s$, which we refer to as *privileged information*. Agent $n$ can observe the behaviours of $O$ as they jointly act in an environment. To achieve **(2)**, we include this representation of $O$ in $o^n$ with probability $p_{\text{obs}}$ at each timestep and linearly anneal $p_{\text{obs}}$ across the $K$ trials from $1 \to 0$. This ensures that the agent learns to act independently by the final trial. Perez et al. [2023] show that increasing opportunities to learn socially leads to less diversity in homogeneous populations, which could be seen as limiting independent discovery in our context, further motivating this constraint. For **(3)**, we add some random noise to the oracle policy according to a tunable parameter $\epsilon$[4]. This encourages the agent to learn that the behaviour of other agents $O$ in the environment may be sub-optimal, in which case agent $n$ should engage in *selective* social learning [Poulin-Dubois and Brosseau-Liard, 2016]. This approach could be seen as imposing an information rate limit [Prystawski et al., 2023] between the oracle and learning agent. We present this process altogether as Algorithm 1, with changes to standard RL[2] in red.

#### 4.1.2 Evaluation

---
**Algorithm 2** In-Context Accumulation During Evaluation

---
1: $\hat{\phi} :=$ init. agent state
2: $\theta :=$ parameters from training
3: $s_0 \sim \hat{S}$ // Sample initial state
4: $\tilde{\phi}_0^{n^*} \leftarrow \hat{\phi}$ // Define an initial reset state
5: **for** each generation $m \in [1, M]$ **do**
6:  **for** each population member $n \in [1, N_{\text{pop}}]$ **do**
7:   $\phi_m^n \leftarrow \hat{\phi}$
8:   $s^n \leftarrow s_0$
9:   **for** each trial $k \in [0, K-1]$ **do**
10:    **if** $k = K - 1$ **then**
11:     $\tilde{\phi}_m^n \leftarrow \phi_m^n$ // Store agent state at beginning of last trial as reset state
12:    **end if**
13:    $p_{\text{obs}} = 1 - k/(K-1)$
14:    $\phi_{m-1}^n \leftarrow \tilde{\phi}_{m-1}^{n^*}$ // Set previous generation state to best reset state from that generation
15:    **while** trial not done **do**
16:     $o_{m-1}^n, o_m^n \sim O(s)$
17:     **if** IsVisible $\sim$ Bernoulli$(p_{\text{obs}})$ **then**
18:      Set previous generation visible in $o_m^n$
19:     **end if**
20:     $a_{m-1}^n \sim \pi_\theta(o_{m-1}^n | \phi_{m-1}^n), a_m^n \sim \pi_\theta(o_m^n | \phi_m^n)$
21:     $\phi_{m-1}^n \leftarrow f_\theta(o_{m-1}^n | \phi_{m-1}^n), \phi_m^n \leftarrow f_\theta(o_m^n | \phi_m^n)$
22:     $s^n, r^n \sim T(s^n, a^n)$
23:    **end while**
24:   **end for**
25:   $n^* = \text{argmax}_{n \in [1, N_{\text{pop}}]} \sum_t \mathbb{I}[k = K-1] r_m^n$ // Select best population member based on final trial performance
26:  **end for**
27: **end for**

---

Having trained agents via Algorithm 1, we evaluate their in-context accumulation. During this evaluation phase, $O$ is replaced by the best[5] of the previous generation $n^*$, as we do not assume

---
[4]For the oracle to capture sub-optimal, but goal-directed behaviour, we inject correlated noise by corrupting the privileged information in $o^0$.

[5]This is an external selection mechanism. We investigate the emergence of selective social learning in Appendix C.

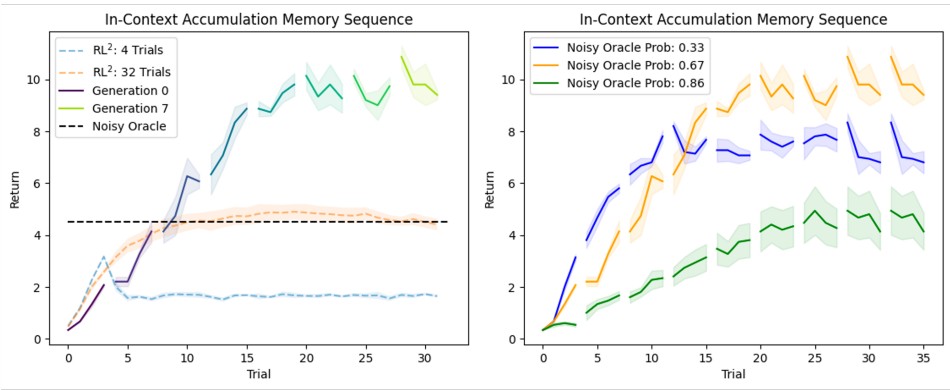

Figure 3: **Left**: In-context accumulation during evaluation on *Memory Sequence*. **Right**: Evaluation results following training with different oracle accuracies.

access to privileged information at test time. This previous best agent, now in *transmission phase*, has its internal state reset between trials. A member of the new generation, in *development phase*, continuously updates its internal state across trials. The same meta-parameters $\theta$ from training are used across generations and only the internal states $\phi$ are used to update each generation. Generations are made up of $N$ agents that interact with $N$ independent environments.

For each generation $g_m$, we store the internal state of each agent on the first step of its final trial and call this the "reset state" $\tilde{\phi}_m^n$. After the final trial, we then keep the previously stored reset state of the best performing agent $\tilde{\phi}_m^{n^*}$ in terms of rewards received in its last trial. When introducing the next generation $g_{m+1}$, generation $g_m$ transitions into transmission phase and we represent the adapted behaviour of the best performing agent by resetting the internal states $\phi_m^{1:N} \leftarrow \tilde{\phi}_m^{n^*}$ at the beginning of each trial. Thus, the next generation $g_{m+1}$ can in essence observe the best of $g_m$ repeating its final trial. In practice, we mask $g_{m+1}$ from the observations of $g_m$ in order to reflect the fact that agents do not expect to see other agents in their final trial, as guaranteed by training attribute **(2)** above. Since the observations of $g_{m+1}$ can include information about the learned behaviours of $g_m$, we implicitly have $g_{m+1} = f(g_m)$, establishing a purely in-context equivalent to generational training (Section 2.4). This process is presented in Algorithm 2.

### 4.2 In-Weights Accumulation

Finally, we present an algorithm for cultural accumulation to take place over the course of training successive generations of policies (i.e., in-weights) rather than through solely updating the internal state (i.e., in-context). This is akin to generational training (Section 2.4), but without modifying the policy-gradient loss, by simply including the previous generation within environments on which the new generation is training, so that their actions can be observed and learned from. This alternative model of cultural accumulation assumes an agent's lifetime to be an entire training run, rather than a single episode. Here, we anneal the probability of observing the previous generation $p_{obs}$ on a given environment timestep linearly over training. The corresponding approach is detailed in Algorithm 3, presented in Appendix A. In-weights RL (i.e., RL training) can be interpreted as updating *procedural* memory [Johnson, 2003] via skill acquisition. We thus interpret in-weights accumulation as gradual, skills-based accumulation. Note that there is no separation between "training" and "evaluation" in this case, because the in-weights algorithm views cultural accumulation as part of the training. This is in contrast to in-context accumulation, which is an evaluation-time algorithm executed by appropriately trained meta-policies. The results that correspond to this setting are in Section 5.2.

## 5 Results

For each of our experiments, we use a Simplified Structured State Space Model (S5) [Smith et al., 2022] modified for RL [Lu et al., 2024] to encode memory, building on the PureJaxRL codebase [Lu et al., 2022a]. This model architecture runs asymptotically faster than Transformers in sequence length and outperforms RNNs in memory tasks [Morad et al., 2024]. PPO [Schulman et al., 2017] is

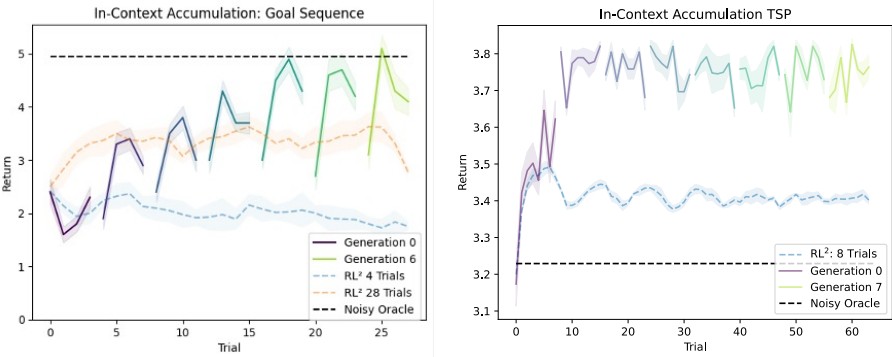

Figure 4: **Left**: In-context accumulation during evaluation on *Goal Sequence*. **Right**: In-context accumulation during evaluation on TSP.

used as the RL training algorithm. For *Goal Sequence* experiments, observations are processed by a CNN, whilst for *Memory Sequence* and TSP, they are processed by feed forward layers. Architectural details are provided in Appendix E and algorithmic hyperparameters in Appendix F. For all in-context experiments, a one-hot encoding of the reward received on each timestep and a one-hot encoding of the current trial number are included each agent's observation. We provide further details on the configuration of each environment in Appendix D. All results are averaged across 10 seeds and the shaded regions of plots show standard error.

## 5.1 In-Context Results

As single-lifetime baselines, we use $RL^2$ trained on: (a) episodes of equivalent length to one generation and (b) episodes of equivalent length to all cumulative generations. All in-context evaluations are performed on held-out environment instances for the same number of trials.

**Memory Sequence:** In Figure 3, we show that in-context learners trained according to Algorithm 1 are capable of accumulating beyond single-lifetime $RL^2$ baselines and even beyond the performance of the noisy oracles with which they were trained when evaluated on a new sequence. Interestingly, when evaluating the accumulation performance of agents trained with oracles of different noise levels, we see that it degrades when oracles are too accurate. This is directly inverse to results in imitation learning [Sasaki and Yamashina, 2021], where the quality of expert trajectories positively impacts the performance achieved when training on these trajectories. We attribute this result to an over-reliance on social learning when oracles are too accurate, which therefore impedes the progress of independent in-context learning during training. Conversely, if oracles are too random, agents do not acquire the necessary social learning skills to effectively make use of prior generations.

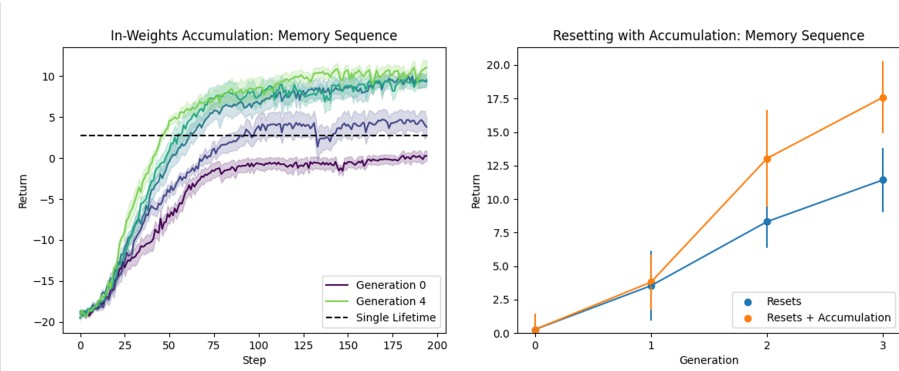

Figure 5: **Left**: In-weights accumulation on *Memory Sequence*. **Right**: In-weights accumulation compounds with resetting. Error bars represent 95% confidence intervals.

**Goal Sequence:** Figure 4 (left) shows that in-context accumulation also significantly outperforms single-lifetime RL$^2$ when evaluated on a new goal sequence. On this more challenging, partially-observable navigation task, we found that higher, but still imperfect oracle accuracies during training produced the most effective accumulating agents. This is likely due to the fact that learning to learn socially in this environment is a much harder problem that involves finding and actively following another agent. We additionally observe that the performance of each generation typically drops slightly on the last trial or two, where agents are entirely or mostly independent. This indicates that although agents are able to independently recall and navigate most of the sequence they have learned, improving on the previous generation, they are still somewhat over-reliant on demonstrations.

**TSP:** We present these results in Figure 4 (right). Again, we see that cultural accumulation enables sustained improvements considerably beyond RL$^2$ over a single continuous context (i.e., lifetime). We only show the RL$^2$ baseline trained on shorter episodes, as the baseline trained on longer episodes failed to make meaningful learning progress. In Figure 2, we show routes traversed by different generations for an example set of city locations. Notably, these routes become more optimised across generations, with later generations exploiting a decreasing subset of edges.

## 5.2 In-Weights Results

**Memory Sequence:** Figure 5 (left) shows that agents can accumulate over the course of training via Algorithm 3. For in-weights accumulation, the sequence is kept fixed over training, meaning that agents are trained to learn as much of a single sequence as possible. After one generation of accumulation, these agents outperform single-lifetime training for a duration equivalent to 5 complete generations. This demonstrates that single-lifetime learners succumb to primacy bias [Nikishin et al., 2022] and converge prematurely. In light of this, we investigate whether in-weights accumulation can be paired with other methods to overcome primacy bias. In particular, we use the simple approach

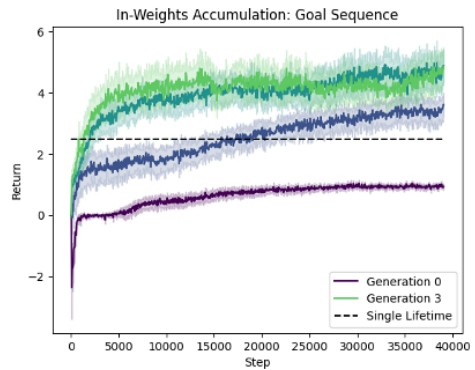

Figure 6: In-weights accumulation on *Goal Sequence*.

of resetting some policy network layers [Nikishin et al., 2022], opting for resetting the final two feed-forward layers, and present these results in Figure 5 (right). We observe that resetting alone helps mitigate premature convergence, but that when we apply resetting to accumulating agents[6], the improvements are compounded and agents reach higher returns than via either method independently.

**Goal Sequence:** Figure 6 shows that in-weights accumulation exceeds the performance of training a single agent for a continuous run equivalent to the total duration of 4 accumulation steps.

**TSP:** We show results for in-weights accumulation in TSP in Figure 7, within Appendix B.

## 6 Related Work

**Social Learning:** Considerable focus has been placed on using social learning to overcome hard exploration problems, imitate human behaviour, and generalise to unseen environments [Ndousse et al., 2021, Bhoopchand et al., 2023, Ha and Jeong, 2023]. Ndousse et al. [2021] trains agents that achieve reasonable performance without a demonstrator by increasing the proportion of episodes with no demonstrator in handpicked increments over training. We employ a similar approach for training agents to eventually act independently by annealing the probability of observing the previous generation across training. To obtain independently capable agents in-context, we anneal the probability of observing the previous generation between trials, which is similar to the expert dropout used in Bhoopchand et al. [2023].

**Machine Learning Models of Cumulative Culture:** A growing body of work uses advances in machine learning to model cultural accumulation *in-silico* [Prystawski et al., 2023, Perez et al., 2023,

---

[6]We initialise the second generation entirely from scratch so that it can learn to learn socially and use resetting with accumulation thereafter.

| | Explicit Imitation Learning | Implicit Third-Person Social Learning |
|---|---|---|
| Non-Generational | Behavioral Cloning [Pomerleau, 1988] Inverse RL [Russell, 1998] | Observational Learning by RL [Borsa et al., 2017] Emergent Social Learning [Ndousse et al., 2021] Learning Few-Shot Imitation [Bhoopchand et al., 2023] |
| Generational Training | Kickstarting [Schmitt et al., 2018] X-Land [Stooke et al., 2022] | **Our Work** |

Table 1: Comparisons to prior work. Generational Training helps overcome local optima to tackle open-ended tasks [Stooke et al., 2022]. Implicit third-person social learning is more general than hand-crafted imitation learning algorithms. Our work combines both to achieve cultural accumulation.

2024]. Perez et al. [2024] use Bayesian RL with constrained inter-generational communication to reproduce social learning processes documented in human populations. This communication is represented in a domain specific language. Rather than requiring an explicit communication channel between agents, we use social learning to facilitate accumulation in a more domain agnostic manner, demonstrating performance gains in multiple distinct settings. Prystawski et al. [2023] demonstrate cultural evolution in populations of Large Language Models, where language-based communication is used as the mechanism for knowledge transfer between generations.

**Generational RL:** The generational process of cultural accumulation can be seen as iterations of *implicit* policy distillation and improvement. Iterative policy distillation [Schmitt et al., 2018, Stooke et al., 2022] is therefore related to our work. Unlike these methods, we do not assume that the new learner has access to the policies of experts or past generations, but only observations of their behaviour as they interact with a shared environment. We also do not explicitly modify the learning objective, leaving the agent free to learn how much or how little to imitate past generations.

## 7 Conclusion

We take inspiration from the widespread success of cumulative culture in nature and investigate cultural accumulation in reinforcement learning. We construct an in-context model of accumulation, which operates on episodic timescales, and an in-weights model, which operates over entire training runs. We define successful cultural accumulation as a generational process that exceeds the performance of independent learning with the same total experience budget. We then present in-context and in-weights algorithms that give rise to successful cultural accumulation on several tasks requiring exploration under partial observability. We find that in-context accumulation can be impeded by training agents with oracles that are either too reliable or too unreliable, highlighting the need to balance social learning and independent discovery. We also show that in-weights accumulation effectively mitigates primacy bias and is further improved by network resets.

**Limitations and Future Work:** An interesting future direction would be to use learned auto-curricula [Dennis et al., 2020, Jiang et al., 2021] for deciding when agents should learn socially or independently, instead of linearly annealing the observation probability. Another extension would be to investigate cumulative culture in settings with different incentive structures, such as heterogeneous or cooperative rewards [Rutherford et al., 2023]. Future work could also study ways we can evolve the process of cultural accumulation itself [Lu et al., 2023a] or learn to influence the learning process of other agents [Lu et al., 2022b, 2023b] for cultural transmission. Finally, we note that whilst understanding cumulative culture has benefits for both machine learning and modelling human society, the consequences of self-improving systems should warrant consideration.

## Acknowledgments

Jonathan Cook is supported by the ESPRC Centre for Doctoral Training in Autonomous Intelligence Machines and Systems EP/S024050/1. Jakob Foerster is partially funded by the UKI grant EP/Y028481/1 (originally selected for funding by the ERC). Jakob Foerster is also supported by the JPMC Research Award and the Amazon Research Award.

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

# A   In-Weights Accumulation

**Algorithm 3** In-Weights Accumulation

1: $\hat{\theta} :=$ init. agent parameters
2: $\tilde{\theta}_0^{n^*} \leftarrow \hat{\theta}$ // Initialise the "random" first generation parameters
3: **for** each generation $m \in [1, G]$ **do**
4:     **for** each population member $n \in [1, N_{\text{pop}}]$ **do**
5:       $B :=$ trajectory buffer
6:       $\theta_m^n \sim \hat{\theta}$
7:       **for** train step $t \in [0, T-1]$ **do**
8:         $p_{\text{obs}} = 1 - t/(T-1)$
9:         $s \sim \hat{S}$
10:         **while** not done **do**
11:           $o_{m-1}^n, o_m^n \sim O(s)$
12:           **if** $p \sim$ Bernoulli$(p_{\text{obs}})$ **then**
13:             Set previous generation visible in $o_m^n$
14:           **end if**
15:           $a_{m-1}^n \sim \pi_{\tilde{\theta}_{m-1}^{n^*}}(o_{m-1}^n)$
16:           $a^n \sim \pi_{\theta_m^n}(o_m^n)$
17:           $s, r \sim T(s, a)$
18:           $B \leftarrow (o_m^n, a_m^n, r_m^n)$ // Append transition to buffer
19:         **end while**
20:         $\theta_m^n \leftarrow \text{update}(\theta_m^n, B)$
21:       **end for**
22:       $n^* = \text{argmax}_{n \in [1, N_{\text{pop}}]} \sum_{t=T-1} r_m^n$ // Select best population member based on final episode performance
23:     **end for**
24: **end for**

# B    In-Weights Accumulation in TSP

In this setting, after two generations, performance exceeds agents trained for a single lifetime equivalent to 5 generations. However, we do not observe as much improvement on subsequent generations.

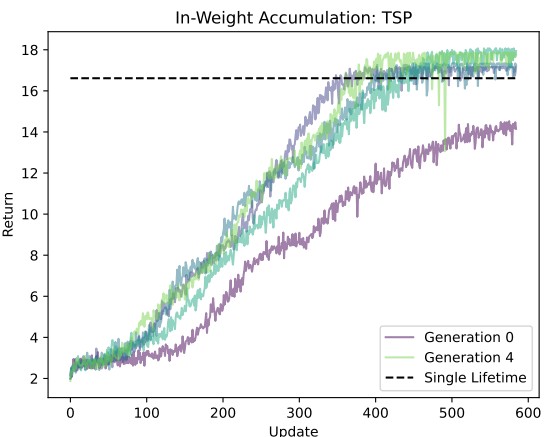

Figure 7: In-weights accumulation on TSP.

## C  Selective Social Learning

In the Goal Sequence experiments, we select the *best performing* agent of the last generation for the current generation of agents to observe during training, automating the selection process. In human and animal cultural accumulation, this selection is instead *learned* through *prestige cues* Barkow et al. [1975], Horner et al. [2010]. Thus, in the Memory Sequence experiments, we do not automatically select the best of the past generation for the agents to observe. Instead, the agents can observe the *entire* last generation. For In-Context Accumulation, we sort the observed oracles and agents by their performance during pre-training and evaluation. Thus, the agents ideally learn to weight the last generation by their relative performance. For In-Weights Accumulation, sorting does not make a difference, as random network initialization is agnostic to the observation ordering. Thus, the agent learns implicitly through observation, which of the past generation is the best to imitate.

# D   Further Environment Details

## D.1   Memory Sequence

For in-context accumulation in Memory Sequence, we use sequences comprised of three digits, with a maximum length of 24, and give the agents four trials per episode. Each generation has a population size of three. We perform initial meta-training for 8e6 timesteps. Agents get a reward of $1.0$ for a correct response and a reward for $-1.0$ for an incorect one. The trial ends when an incorrect response is given or the maximum length is reached.

For in-weight accumulation in Memory Sequence we use sequences comprising of ten digits, with a maximum length of 24. Agents train for 1e5 timesteps. Each generation has a population size of five. Agents get a reward of $1.0$ for a correct response and a reward for $-1.0$ for an incorect one. The episode ends when the maximum length is achieved.

## D.2   Goal Sequence

We use $7 \times 7$ grids with 3 goal types. Agents observe a $3 \times 3$ grid of cells directly in front of them. Observations are symbolic with 4 channels (one for each goal type and one for agents). On each timestep, an agent can take one of three actions: `move forward`, `turn left`, `turn right`. Agents receive $+1$ reward for hitting the correct next goal and $-1$ reward for hitting the incorrect next goal. For in-context experiments, trials are 30 steps long and there are 4 trials in an episode. For in-weights experiments, episodes are 50 steps long.

## D.3   TSP

For in-context accumulation in TSP, we use six cities and provide eight trials per episode. The city positions are uniformly sampled in the unit square. Each generation has a population size of three. We perform initial meta-training for 8e6 timesteps. Agents get a reward of $\frac{(\sqrt{2}-\text{dist}(\text{cur city},\text{next city}))}{\sqrt{2}}$ if the selected next city is valid. Agents get a reward of $-1.0$ and the trial ends if they select an already-visited city.

For in-weight accumulation in TSP, we use twenty-four cities. Each generation has a population size of eight. Agents train for 3e5 timesteps. The reward setup is equivalent to above.

# E  Architecture Details

## E.1  *Memory Sequence* **and TSP**

The policy and value networks share two fully layers and a GRU layer. Values and policies are computed with two fully connected layers. All layers use leaky ReLU activations. Here, we take the example of a 4-dimensional input for the 4-dimensional embedding of an action observation and assume the in-weights setting, so no trials. The final output is 10-dimensional in the case of a random sequence made up of 10 distinct digits.

- Shared input layers:
    - FC (4, 128)
    - FC (128, 256)
    - FC (256, 256)
    - GRU (256, 256)
- Value MLP:
    - FC (256, 128)
    - FC (128, 128)
    - FC (128, 1)
- Policy MLP:
    - FC (256, 128)
    - FC (128, 128)
    - FC (128, 10)

## E.2  *Goal Sequence*

The policy and value networks share three convolutional layers, a fully connected layer, and an S5 layer. Values and policies are computed with two fully connected layers. Convolutions use leaky ReLU activation functions and all other layers use tanh activation functions. Inputs are 5-dimensional for one-hot encodings of 3 goal types, walls and agents. A 3-dimensional vector for one-hot encodings of reward is concatenated to the output of the last convolutional layer. A one-hot encoding of the trial would also be included in the in-context setting.

- Shared input layers:
    - Conv (5, 32), $1 \times 1$ filters, stride 1, padding 0
    - Conv (32, 64), $1 \times 1$ filters, stride 1, padding 0
    - Conv (64, 64), $1 \times 1$ filters, stride 1, padding 0
    - FC (576 + 3, 256)
    - S5 (256, 256)
- Value MLP:
    - FC (256, 64)
    - FC (64, 64)
    - FC (64, 1)
- Policy MLP:
    - FC (256, 64)
    - FC (64, 64)
    - FC (64, 3)

# F  Hyperparameters

## F.1  *Memory Sequence* and TSP

| | |
|---|---|
| population size | 5 |
| learning rate | $2.5 \times 10^{-5}$ |
| batch size | 4 |
| rollout length | 128 |
| update epochs | 4 |
| minibatches | 4 |
| $\gamma$ | 0.99 |
| $\lambda_{GAE}$ | 0.95 |
| $\epsilon$ clip | 0.2 |
| entropy coefficient | 0.01 |
| value coefficient | 0.5 |
| max gradient norm | 0.5 |
| anneal learning rate | False |

## F.2  *Goal Sequence*

| | |
|---|---|
| population size | 4 |
| learning rate | $1 \times 10^{-5}$ |
| batch size | 128 |
| rollout length | 32 |
| update epochs | 8 |
| minibatches | 8 |
| $\gamma$ | 0.99 |
| $\lambda_{GAE}$ | 0.95 |
| $\epsilon$ clip | 0.2 |
| entropy coefficient | 0.01 |
| value coefficient | 0.5 |
| max gradient norm | 0.5 |
| anneal learning rate | False |

# G    Compute Resources

*Memory Sequence* and TSP experiments were run on a single NVIDIA RTX A40 GPU (40GB memory) in under 20 minutes. Training of in-context learners in *Goal Sequence* was run in under 8 minutes on 4 A40s (oracle training runs in the same time). In-weights accumulation in *Goal Sequence* was run in 30 minutes on 4 A40s.

