# OpenReview forum: "Artificial Generational Intelligence: Cultural Accumulation in Reinforcement Learning"
_NeurIPS.cc/2024/Conference — NeurIPS 2024 poster_

### Official Review · Reviewer_R6qB · 2024-07-06

**Soundness:** 2
**Presentation:** 2
**Contribution:** 3
**Rating:** 6
**Confidence:** 4

**Summary:**

The authors introduce two methods for learning agents to trade-off imitation and exploration across generations, by incorporating the behavior of noisy oracles and/or the best-performing agents in prior generations into the observations of the next generation of agents. Two settings are studied: in-context learning and in-weights learning.  The in-context learning involves:
- a training phase, in which a meta-RL algorithm learns a recurrent RL algorithm that uses observations from a noisy oracle to learn to solve a POMDP by updating a hidden state
- a test phase in which the RL algorithm is frozen but the hidden state can still change, and the oracle is switched out with the agent with the highest performing hidden state from the previous generation.

The in-weights accumulation has no train-test split or hidden state, and just trains the RL algorithm from scratch at each generation, allowing it to observe both the oracle and the best agent from the previous generation. Noise is added to the oracle and the probability of observing it is decreased over time, so the agents must learn to explore and act independently.
They demonstrate that their methods outperform non-cumulative/single life baselines in three simple partially observable tasks.

**Strengths:**

- The approach is novel and interesting
- The work is well situated among related prior work in generational methods and social learning.

**Weaknesses:**

- The writing about the algorithms is quite unclear, especially the training algorithm of the in-context accumulation and the in-weights accumulation algorithm are only partially explained in the main text while the pseudocode is relegated to the appendix (see questions section) the paper could benefit from more thorough explanations of the methods in the main text

- The ability to observe an oracle with access to privileged information (even with the addition of noise) seems like an unrealistic choice, artificially making the problem easier especially in the in-weights setting, since the oracle can be observed throughout rather than only at training time. Even worse, the method is highly sensitive to how much noise is added to the oracle, and the optimal noise amount also varies significantly across settings. Why can’t the agents in this approach learn to learn robustly from prior generations without an oracle?

- There is only one baseline used in each setting, and it is unclear how much of an even playing field they’re tested on. How much were they tuned, e.g. by adjusting the learning rate decay if premature convergence was a major factor? Could they access the oracle demonstrations in any way, or could only the accumulation algorithms observe the oracle? Equation 1 does not account for the population size N, just the number of generations- shouldn’t the baseline’s experience budget be multiplied by another factor of N?

- Cultural accumulation is not just passing down information about the unobserved state, but can also involve passing down procedural knowledge (e.g. mathematics or dance). the authors claim that in-weights accumulation can be interpreted as updating procedural memory, but the environments they test in do not seem to have the right type of complexity to demonstrate this. It would help if the authors tested in a setting that is fully observable but with more complex dynamics requiring the development of skills, such as half-cheetah

- Only observing the best performing agent from the previous generation seems unnaturally limited - humans get to observe a range of behaviors from other humans and may learn from the mistakes of those who perform poorly, or could combine together two suboptimal agents’ strategies to make an optimal one.

**Questions:**

- How exactly is the oracle’s behavior incorporated into the observation?

- How is a^{-i} used by the in-weights algorithm?

- How sensitive is the approach to the population size?

**Limitations:**

yes

---

> ### Author Rebuttal · Authors · 2024-08-06
>
> We would like to thank the reviewer for their clear and focused review. We are pleased that the reviewer finds that “the approach is novel and interesting” and that “the work is well situated among related prior work in generational methods and social learning.”
>
> The reviewer seems to have grasped the core contributions of our work, however there is an important mistake made in their summary. They claim that during in-weights accumulation we allow new generations “to observe both the oracle and the best agent from the previous generation,” however this is distinctly false as we make no use of an oracle for in-weights accumulation. The oracle is solely used for training social learning abilities ahead of *in-context* accumulation.
>
> # Points of clarification
>
> > The paper could benefit from more thorough explanations of the methods in the main text.
>
> We thank the reviewer for pointing this out and agree to bring further algorithmic details into the main text.
>
> > The ability to observe an oracle with access to privileged information seems like an unrealistic choice, especially in the in-weights setting.
>
> We clarify again that we make no use of oracles in the in-weights setting. In conjunction with the point above, we appreciate that further details on the in-weights algorithm in the main text would be of value in preventing misunderstandings such as this.
>
> As the reviewer indicates, agents can learn to learn from prior generations when training (i.e., in-weights), but this is not possible from scratch during purely in-context accumulation, which is what necessitates the use of an oracle during training on the train subset of environments. We would like to note that this is a less restrictive assumption than access to expert agents on test environments [1, 2].
>
> > The method is highly sensitive to how much noise is added to the oracle.
>
> This work is meant as an investigation into cultural accumulation in RL and so we perceive insights such as this to be of immense value; to the best of our knowledge, no existing work has revealed how social learning in different settings (i.e., in environments shared with agents of different abilities) impacts downstream cultural accumulation. This result is counterintuitive when compared, at face value, to the need for high quality expert trajectories in imitation learning [3].
>
> > How much were the baselines tuned, e.g. by adjusting the learning rate decay if premature convergence was a major factor?
>
> The baselines use the same underlying RL algorithm and have the same hyperparameters. The best hyperparameters found for baselines and accumulation were the same.
>
> > Could the baselines access the oracle demonstrations in any way, or could only the accumulation algorithms observe the oracle?
>
> For in-context accumulation, we tried training the RL^2 baselines with an oracle, but since oracles are not seen at test time, they in fact then do worse.
>
> > Equation 1 does not account for the population size N, just the number of generations- shouldn’t the baseline’s experience budget be multiplied by another factor of N?
>
> Populations train in parallel in independent environments for baselines and the main algorithm. Because of this equivalence, N would cancel out on the left and right sides of the equation.
>
> # On environments
>
> > Cultural accumulation is not just passing down information about the unobserved state, but can also involve passing down procedural knowledge [...] It would help if the authors tested in a setting that is fully observable but with more complex dynamics.
>
> While it's true that many people who use the term 'procedural' in neuroscience are interested in continuous motor control, it's certainly not the case that neuroscientists intend the term only to apply in the continuous case. The term is also used in the literature on the neuroscience of multiple memory systems [4], which is what inspires our in-context/ in-weights distinction.
>
>
> To address the reviewer’s request for results on a more complex, fully-observable environment, we ran in-weights accumulation on MinAtar Breakout and Freeway, and Half-Cheetah as requested, and yielded results in Figure 1 and Figure 2 of the attached pdf document.
>
>
> > Only observing the best performing agent from the previous generation seems unnaturally limited.
>
>
> We in fact have positive results addressing this exact concern. Appendix D details how the results on Memory Sequence assess the selective social learning described by the reviewer. We excluded this discussion from the main text due to space constraints, but will ensure some of it is included in the main body of a camera-ready version.
>
>
> # Addressing questions
>
> 1. This is done simply through being in a shared environment. For Goal Sequence, this corresponds to observing the other agent whenever it enters the field of view. For Memory Sequence and TSP, this corresponds to simply observing the last action taken by the other agent.
>
> 2. Inferences about a^{-i} (either from direct observation or partially observed state transitions depending on the constraints of the environment) can implicitly be used by the learning agent to inform its own actions. There is no explicit use of a^{-i}, which is the unique quality of social learning and what allows learners to flexibly balance social learning and independent learning to achieve generational accumulation.
>
> 3. For all in-context experiments, the population size is only three. For in-weights experiments, the population size is only five. The consistency of results within a population can further be seen by the reasonably small standard error regions in our plots.
>
> [1] Emergent social learning via multi-agent reinforcement learning, Kamal Ndousse, et. al, 2021
>
> [2] Learning few-shot imitation as cultural transmission, Bhoopchand, et. al, 2023
>
> [3] Behavioural cloning from noisy demonstrations, Fumihiro Sasaki, 2021
>
> [4] Memory systems of the brain: a brief history and current perspective, Larry R. Squire, 2004

---

> > ### Comment · Reviewer_R6qB · 2024-08-08
> > **thank you for the response**
> >
> > Thank you for answering my questions and addressing many of my concerns with the paper. I still have some remaining questions and concerns:
> >
> > >we make no use of an oracle for in-weights accumulation
> >
> > Line 13 of "**Algorithm 3** In Weights-Accumulation" says "Set oracle visible in $o^i$" - so I guess this must be a typo?
> >
> > **Clarifications**
> >
> > > Populations train in parallel in independent environments for baselines and the main algorithm.
> >
> > Are you saying that for population size N, you train N different seeds for the baseline model and select the best one? I can't see this specified anywhere. I'm also just noticing now that there are no error bars shown for the single lifetime baselines- is the value plotted in the horizontal dashed line the highest return ever achieved?
> >
> > > The baselines use the same underlying RL algorithm and have the same hyperparameters. The best hyperparameters found for baselines and accumulation were the same.
> >
> > This does not actually answer my question: I asked how much you tried tuning the baseline's hyperparameters, and specifically if you tuned the learning rate decay. Are you implying that you did just as much tuning for both the baseline and the accumulation? I would be satisfied with that, if there was specific attention paid to tuning of the learning rate and learning rate schedules. My underlying concern is that cultural accumulation could be no more effective than a well chosen learning rate schedule (especially one with warm restarts, e.g. [1]), if the main issue with the baselines is premature convergence.
> >
> > > Appendix D details how the results on Memory Sequence assess the selective social learning described by the reviewer.
> >
> > I don't see this in appendix D? It doesn't comment on results, just environment details
> >
> > **On Environments**
> >
> > Thank you for sharing results on the environments involving more complex control. It is encouraging to see that each generation seems to outperform the last, except for breakout MinAtar where generations 1-3 appear to perform the same (why do you think it is failing there?) could you please also show the error bars and the single lifetime baseline performances in your plots?
> >
> > Loshchilov, Ilya, and Frank Hutter. "Sgdr: Stochastic gradient descent with warm restarts." arXiv preprint arXiv:1608.03983 (2016).

---

> ### Author Response · Authors · 2024-08-08
> **Taking on feedback and addressing further concerns**
>
> Thank you for taking the time to read our rebuttal and for sharing your further questions. We endeavour to address these in this comment.
>
> # Further clarifications
>
> In addition to providing the following clarifications, we will update our manuscript to explicitly state them.
>
> > Line 13 of "Algorithm 3 In Weights-Accumulation" says "Set oracle visible in $o^{i_n}$" - so I guess this must be a typo?
>
> Thank you very much for identifying this! Yes, this can be considered a typo, here we simply mean the previous generation by 'oracle'. We will adjust our manuscript to reflect this and avoid further confusion.
>
> > Are you saying that for population size N, you train N different seeds for the baseline model and select the best one? [...] Is the value plotted in the horizontal dashed line the highest return ever achieved?
>
> This is correct and thank you for pointing out that this has not been explicitly stated. Taking the argmax over seeds is also the reason that there is no error region on the single-lifetime in-weights baselines.
>
> > Are you implying that you did just as much tuning for both the baseline and the accumulation? [...] if there was specific attention paid to tuning of the learning rate and learning rate schedules...
>
> We indeed carefully tuned learning rate schedules for the single-lifetime in-weights baselines. As mentioned, we found that tuning hyperparameters for single-lifetime baselines also contributed to improved performance for accumulation (i.e., performance improvements sustained over more generations), including the tuning of learning rate decay. Given that this has emerged as important information to the reviewer, we can include these sweeps in the Appendix of a camera-ready version of the manuscript.
>
> We would also like to state that in-context accumulation overcomes an entirely different issue to premature convergence, one of effectively exploring a new environment over long contexts.
>
> > I don't see this in appendix D? It doesn't comment on results, just environment details.
>
> We sincerely apologise for this oversight. In our updated manuscript, we have included this information in the Appendix but it was not present in the original submission. We provide the contents at the end of this comment.
>
> > ...except for breakout MinAtar where generations 1-3 appear to perform the same...
>
> Generation 3's final performance is 50.8 whereas generation 2's final performance is 44.3, but we concede that generations 1 and 2 are approximately the same.
>
> > Could you please also show the error bars and the single lifetime baseline performances in your plots?
>
> We are unable to upload new figures or update the pdf during the discussion period, so we provide the baselines (once again as a maximum over the same population size) numerically here and provide the numeric maximum difference in returns between seeds (over the training of all generations) to provide some information regarding error bars.
>
> | Task                           | Single-Lifetime Baseline | Maximum Return Range |
> |-------------------------|-----------------------------|------------------------------|
> | Minatar: Breakout      | 36.5                                | 8.2                                    |
> | Minatar: Freeway       | 39.1                                | 3.2                                    |
> | Half-Cheetah             | 4773.8                            | 219.6                                |
>
> # Appendix on selective social learning
>
> In the Goal Sequence experiments, we select the best performing agent of the last generation for the
> current generation of agents to observe during training, automating the selection process. In human
> and animal cultural accumulation, this selection is instead learned through prestige cues Barkow et al.
> [1975], Horner et al. [2010]. Thus, in the Memory Sequence experiments, we do not automatically
> select the best of the past generation for the agents to observe. Instead, the agents can observe the
> entire last generation.
>
> Jerome H Barkow, Akinsola A Akiwowo, Tushar K Barua, MRA Chance, Eliot D Chapple,
> Gouranga P Chattopadhyay, Daniel G Freedman, WR Geddes, BB Goswami, PAC Isichei, et al.
> Prestige and culture: a biosocial interpretation [and comments and replies]. Current Anthropology,
> 16(4):553–572, 1975.
>
> Victoria Horner, Darby Proctor, Kristin E Bonnie, Andrew Whiten, and Frans BM de Waal. Prestige
> affects cultural learning in chimpanzees. PloS one, 5(5):e10625, 2010.

---

> ### Comment · Reviewer_R6qB · 2024-08-09
> **thank you for further clarifications**
>
> Regarding Breakout MinAtar, given the amount of fluctuation in the return I do not find it convincing that generation 3 performed better than previous generations just because the return at step 1200 is higher (it appears to dip below generation 1 less than a hundred steps earlier).
>
> I am satisfied with the authors' other replies. I believe the clarity of the paper will be greatly improved if the authors make the changes they describe in their response, and my major concerns have been addressed, thus I will increase my score.

---

> > ### Author Response · Authors · 2024-08-11
> > **Thank you**
> >
> > Thank you for bringing to light points that will improve the paper's clarity and for raising your score. We greatly appreciate the time spent reviewing the paper and engaging in this discussion.

---

### Official Review · Reviewer_h1KJ · 2024-07-09

**Soundness:** 4
**Presentation:** 2
**Contribution:** 3
**Rating:** 7
**Confidence:** 3

**Summary:**

This paper introduces the problem of modelling cultural accumulation in populations of deep RL agents, with evolution happening through non-communicative social learning. The paper introduces 2 setups for cultural accumulation: one where the agents learn in-context from other agents within an episode composed of several trials through meta-RL (called the in-context setup), and one where the agents learn from other agents whose behavior is sometimes visible in the agent's observations (called the in-weight setup).

In the in-context setup, agents are first trained alongside a noisy oracle (with access to the oracle annealed to 0 over the course of the episode) but imitating the oracle is not enforced in the loss; social learning is implicit. The oracle is kept noisy to encourage agents to learn on their own, so that cultural accumulation can occur. At evaluation time oracles are replaced by the previous generation of agents. There is no separate phase for training in the in-weights case.

The paper tests algorithms for cultural accumulation in three experimental environments. The first is a sequence memorization task, the second is a goal sequence memorization task in a gridworld with egocentric view, and the third a multi-agent, partially-observable traveling salesperson problem. Baselines are RL$^2$ algorithms with no social learning (single lifetime) for the in-context setup, and single lifetime RL training for the in-weight setup. The paper shows consistent improvement on all tasks when cultural accumulation happens with respect to single-lifetime baselines.

**Strengths:**

* The questions being studied are worthwhile and timely; as our understanding of the role of cultural evolution in humanity's success improves it is important that AI researchers investigate related questions with artificial agents.

* The positioning with respect to the existing literature is good and differences between this setup and existing works studying cultural evolution in agent populations are highlighted.

* The background section does a good job of refreshing readers' minds with respect to POMDPs, meta-RL and generational learning;

* The experiments and their presentation are of excellent quality with adequate error bars and offsets, and support the hypothesis that cultural accumulation allows populations of agents to reach better policies;

* The results are significant, and I can see other researchers being inspired by this approach.

**Weaknesses:**

* The paper could be clearer: the fact that there are multiple approaches on multiple environments makes the paper complex and difficult to parse. Maybe the in-weights results could go in the appendix and more space could be devoted in the main text to giving examples or longer explanations of the tasks, as well as of the meta-RL setup? I think that some notations might need polishing, see Questions section below.

* The paper distinguishes itself from [Perez et al 2023, 2024] by adopting a model of cultural evolution based only on imitation of non-pedagogic behavior, i.e. agents imitating other agents doing the same task. While this is a completely valid form of social learning and definitely present in human learning, the first introduction paragraph suggests that this is the main mechanism leading to humanity's success as a species. This is not the case, and it is known that language as a means for instruction and rapid communication of knowledge, explicit teaching, and shared norms and institutions are an important (and human-specific, whereas imitation of behavior occurs in great apes) component of cultural evolution (see Henrich 2019 as well as Tomasello 2019). This should be discussed in the paper.

**Questions:**

* What is the deep RL algorithm underlying both the in-context and the in-weights setup? I am not sure this is mentioned in the main body of the paper;

* What is the extent to which these results could serve as a model for human evolution, knowing how much RL agents and humans differ, and how different the current setup is compared to human learning setups? How do such experiments compare, as far as modelling of human evolution is concerned, to simple models from the cultural evolution literature like those of [Enquist et al, 2008] (or of [Fogarty et al 2015] in innovation)? Could more realistic experiments be devised?

* Couldn't there have been simpler ways to model cultural evolution using LLMs instead of RL agents, since they already have meta-learning capabilities? (for instance, LLM agents with skill repertoires like LMA3 [Colas et al 2023] or Voyager [Wang et al 2023]).

* (minor) L181 the claim about populations should say whether this is a simulation or human result;

* Why does the TSP agent not benefit that much from more than 1 transmission event, do you think? Is there an exploration problem?

* Why is cultural accumulation better than single lifetime for these tasks? Is there a plasticity problem with the single-lifetime agent? In evolution a part of the advantage of generational transmission is that environments are changing and populations need to be adaptable. There are also aging processes and the difficulty of maintaining lifeforms for arbitrary amounts of time. This is not the case in simulated agents for fixed environments, so how come cultural accumulation provides an advantage in this setting? Are there differences in the amount of exploration in the generational vs single lifetime experiments?

### Notations
(minor)

* l68 what is the observation space? shouldn't the observation function be defined on $O: \mathcal{O} \times S \rightarrow [0, 1]$, defining a joint distribution on observations and states (assuming that $\mathcal{O}$ is the observation space)?

* The $g_m$ notation is not clear when first encountered, are these model weights?

* l70 do the parameters depend on the time t?

* l105 Not sure I get this notation: $g_{m+1} = f (·|g_m)$, is $g_m$ a distribution? Why not just $f(g_m)$?

**Limitations:**

* The paper is concerned with tasks where a clear reward function is defined beforehand. While this is important and models some aspects of tasks early humans had to face, they are not open ended and not subject to variation. Studies of cultural evolution in simulation must eventually tackle open-ended domains, and even better, domains where the population of agents themselves alter the task landscape as they evolve (see for instance how language or the invention of water containers has influenced human evolution).

* The skills learned by agents are not compositional, and thus do not model the stepping-stone aspect of human cultural evolution that allows populations to build upon existing sets of knowledge or combine them to create larger cultural adaptations (the way the Voyager agent implements skill compositionlity by writing programs could be a source of inspiration here);

* The exploration/innovation aspect of cultural evolution is under-represented in this paper. Could exploratory behavior also be meta-learned as it leads to higher long-term innovation and thus reward? Are the test environments complex and variable enough to test this?

* The in-context setup requires access to an oracle to steer agents towards learning to imitate other agents behavior. I acknowledge that this paper does not claim to model the emergence of social learning during evolution, but it would have been significant to get the agents to imitate one another without access to optimal behavior on the task.

---

> ### Author Rebuttal · Authors · 2024-08-06
>
> We would like to thank the reviewer for their extensive and detailed review. We are pleased that the reviewer finds that “the questions being studied are worthwhile and timely”, “the positioning with respect to the existing literature is good”, “the background section does a good job of refreshing readers' minds”, “the experiments and their presentation are of excellent quality” and they “can see other researchers being inspired by this approach”.
>
> In their summary, the reviewer demonstrates a deep and thorough understanding of the contents of the paper, and we would like to thank them for this attention to detail.
>
> We appreciate the reviewer’s constructive feedback and would like to address the weaknesses they state below.
>
> # On clarity
>
> > The paper could be clearer: the fact that there are multiple approaches on multiple environments makes the paper complex and difficult to parse.
>
> Whilst we consider investigating both the in-context/ in-weights settings to be an important contribution, especially as in-context learning becomes an increasingly used paradigm, we acknowledge that there is a lot for the reader to keep in mind. Moving results for some environments, and descriptions of the corresponding environments, to the appendix could be an appropriate solution that would also enable us to include more detail on the algorithms themselves in the main text.
>
> # On the setting
>
> > Language as a means for instruction and rapid communication [...] should be discussed in the paper.
>
> We thank the reviewer for pointing this out and emphatically agree that language is a core transmission mechanism underpinning cultural accumulation in humans. We agree that acknowledging this in the paper would add clarity to that portion of the introduction.
>
> # Addressing questions
>
> 1. PPO. We apologise for this being missing in the main text and will ensure to include it there.
>
> 2. The Memory Sequence environment is actually lifted directly from a controlled study of human cultural accumulation [1]. We believe that settings such as this, which allow for investigating the core mechanisms of the focal phenomenon without additional complexity, are useful for an initial work of this kind. Some verification of this work’s utility as a model of human cultural accumulation is shown by the emergent impact of the “skill level” of other agents on social learning and accumulation, which is observed among humans [2]. The relevance of the in-context and in-weights distinction to multiple-memory systems [3] and the role this plays in cultural accumulation [4] is also of value. Scaling to settings that more closely resemble cultural accumulation “in the wild” is important future work.
>
> 3. Meta-learning arises analogously in both LLM pretraining and RL^2-style meta-RL; we call it “in-context” to be explicit about that parallel. In this work, we focused on  tabula-rasa, communication-free cultural accumulation, but see work leveraging LLMs as an important direction. As a preliminary step in this direction, we ran an analogous experiment to in-context accumulation on Memory Sequence with GPT-4 and show these results in Figure 3 of the attached pdf document.
>
> 4. Acknowledged. This is firstly a human result that is then demonstrated in artificial agents in the cited paper.
>
> 5. We believe this is likely an exploration challenge as the reviewer suggests.
>
> 6. Yes, we verify that there is a plasticity problem in the in-weights setting by showing that this is partially mitigated by the common approach to dealing with plasticity loss of partial network resets. Importantly, partial resets compound with in-weights accumulation when used together, as Figure 5 (right) shows. In the in-context setting, we expect that accumulation is an effective way of overcoming the challenges of learning across long contexts.
>
> # Addressing notation
>
> 1. We thank the reviewer for picking up on this, their definition is correct.
>
> 2. These are model weights (in-weight) or hidden state (in-context). We will make this clearer in a camera ready version.
>
> 3. No, that is a typo and we thank the reviewer for pointing this out.
>
> 4. You are correct that this interpretation is unintended and we have remedied this by updating our manuscript.
>
> # Addressing limitations
>
> 1. We agree that this would be an exciting direction and indicate this in Section 7.
>
> 2. In a simple way, they are compositional in that agents can build knowledge of the environment, and corresponding navigational routes in Goal Sequence and TSP, across generations. We agree that Voyager-like skill composition, or better yet a persistent environment would be exciting directions to consider for cultural accumulation in learning agents.
>
> 3. The exploration-imitation tradeoff is indeed being meta-learned (in the in-context setting) as agents are learning to balance these as they adapt in-context to new environments. The hard exploration component of these tasks (as shown by lower single-lifetime RL performance) is what enables us to study this.
>
> [1] Sequence memory constraints give rise to language-like structure through iterated learning, Hannah Cornish, et. al, 2017
>
> [2] Cumulative cultural learning: Development and diversity, Cristine H. Legare, 2017
>
> [3] Memory systems of the brain: a brief history and current perspective, Larry R. Squire, 2004
>
> [4] Multiple timescales of learning indicated by changes in evidence-accumulation processes during perceptual decision-making, Aaron Cochrane, et. al, 2023

---

> > ### Comment · Reviewer_h1KJ · 2024-08-08
> >
> > I am glad that you took the time to answer my questions and incorporate my comments. I hope the paper is accepted and am looking forward to see follow-up work in more complex transmission setups.

---

> > > ### Author Response · Authors · 2024-08-09
> > >
> > > Thank you for your kind response. We humbly ask if you will consider raising your confidence rating and/or score in light of having addressed your questions and comments. Equally, if there is any further information we can provide, please do let us know.

---

### Official Review · Reviewer_9itL · 2024-07-10

**Soundness:** 4
**Presentation:** 3
**Contribution:** 4
**Rating:** 8
**Confidence:** 4

**Summary:**

The paper "Artificial Generational Intelligence: Cultural Accumulation in Reinforcement Learning" introduces the concept of cultural accumulation in reinforcement learning (RL), where agents benefit not only from their own experiences but also from the knowledge passed down from previous generations, akin to human cultural evolution. The authors propose two models for this accumulation: in-context accumulation, which occurs within single episodes, allowing fast adaptation, and in-weights accumulation, which embeds knowledge in neural network weights over longer training periods. Through experiments on tasks such as Memory Sequence, Goal Sequence, and the Traveling Salesperson Problem (TSP), the paper demonstrates that agents trained with cultural accumulation outperform those trained without it. This work represents the first demonstration of general models achieving emergent cultural accumulation in RL, opening new directions for more open-ended learning systems and providing fresh insights into modelling human cultural processes with artificial agents.

**Strengths:**

1. Innovative Concept: The introduction of cultural accumulation in RL is novel and aligns well with how learning and knowledge transfer occur in human societies.
2. Robust Experimental Design: The experiments are well-designed to test the hypotheses, with clear evidence showing the benefits of cultural accumulation.
3. Comprehensive Analysis: The paper provides a thorough analysis of both in-context and in-weights accumulation, exploring different environments and scenarios.

**Weaknesses:**

1. Complexity of Models: The proposed models may be complex to implement and require significant computational resources, which might limit their applicability.
2. Scalability Concerns: While the models work well in the presented tasks, it is unclear how they will scale to more complex or real-world scenarios.
3. Limited Real-World Applications: The paper primarily focuses on theoretical and controlled environments. More discussion on potential real-world applications and implications would strengthen the paper.

**Questions:**

1. How do you anticipate the scalability of your models to more complex or real-world tasks beyond the experimental environments used in this paper?
2. What are the computational resources required to train these models, and how do they compare to traditional RL methods?
3. How do you determine the optimal balance between social learning and independent discovery during training?
4. How do your models handle diverse or rapidly changing environments where cultural knowledge might quickly become outdated?

**Limitations:**

1. While the paper demonstrates the models on specific tasks, there is limited discussion on the scalability and complexity of the models in more complex or real-world scenarios.
2. The authors do not provide a detailed analysis of the computational resources required for their models, which could be a limitation for practical applications.
3. The potential real-world applications and implications of the proposed models are not thoroughly explored. Discussing these aspects would provide a clearer picture of the practical relevance of the work.

---

> ### Author Rebuttal · Authors · 2024-08-06
>
> We would like to thank the reviewer for their positive and informative review. We are pleased that the reviewer finds that the paper exhibits an “innovative concept”, “robust experimental design” and “comprehensive analysis”. Whilst we appreciate the reviewer’s acknowledgement of these strengths, we would like to address the claimed weaknesses.
>
> # On complexity
>
> > The proposed models may be complex to implement and require significant computational resources, which might limit their applicability.
>
> We in fact make very minimal changes to existing RL algorithms. For evidence of this, see Algorithm 2 (in-context accumulation training), which has 3 simple lines of difference relative to vanilla RL^2. The core approach of in-weights accumulation simply involves training agents in successive generations and keeping around the previous generation in a shared environment, rather than using explicit policy distillation as in [1].
>
> A distinct advantage of cultural evolution is that it more efficiently explores the space of behaviours than genetic algorithms, which rely on cumulative mutations in genotype (e.g., network weights). This means that our implementations of cultural accumulation in RL require only very limited compute resources. We detail the compute resources used in Appendix H. All experiments can be run on a single 40 GB A40 GPU in under 1 hour thanks to our JAX-based algorithm and environment implementations. We cut this down to a maximum train time of 8 minutes by using 4 GPUs for Goal Sequence experiments.
>
> # On scalability
>
> > While the models work well in the presented tasks, it is unclear how they will scale to more complex or real-world scenarios.
>
> We view this work as an important first step in a new direction, as recognised by the acknowledgements of novelty by the reviewer as well as reviewers h1KJ and R6qB. As such, we begin by seeking to demonstrate and understand cultural accumulation in RL using environments that are amenable to doing so. As a first step in considering additional environments, we present results for in-weights accumulation on Atari Breakout and Freeway in Figure 1 of the attached pdf document, and Brax Half-Cheetah in Figure 2.
>
> We also believe that the use of language models in addition to RL is a natural direction for scaling up this work, but one that we think warrants its own paper.
>
> # On real-world applications
>
> > More discussion on potential real-world applications and implications would strengthen the paper.
>
> One application is modelling human cultural accumulation “in-silico”, which we believe is a key contribution of this work. To the best of our knowledge, we are the first to show the effect of demonstrator ability (i.e., oracle noise) on emergent cultural accumulation, an effect also documented amongst humans [2].
>
> We do however acknowledge that more discussion of the real-world applications that might arise from this new line of work would be of value to the paper. One can imagine teams of agents at deployment time figuring out how to improve on solutions to tasks e.g. in a warehouse order fulfilment setting, in a home assistant setting, in a delivery drone setting etc. Our setup is particularly amenable to putting a human in the loop too, because in-context accumulation only requires observational data of the human.
>
> Finally, in the era of large scale pretraining in AI, it is useful to study the mechanisms by which learning systems can learn from and improve upon data representing other agents.
>
> # Addressing questions
>
> 1. See “On scalability” and “On real-world applications” above.
>
> 2. See “On complexity” above.
>
> 3. The agent discovers this entirely through learning to balance imitation and independent learning in the in-weights setting. In the in-context setting, we find that oracle noise level will bias it in one direction or another and therefore needs to be considered. As with other hyperparameters, the best way to discover the optimal oracle noise level is via empirical search.
>
> 4. We would expect that independent exploration would be the most effective strategy in this case, depending on the learning sample efficiency of the task at hand. Here cultural accumulation should excel relative to imitation learning or policy distillation because the social vs asocial tradeoff is learned online.
>
>  [1] Open-ended learning leads to generally capable agents, Stooke, et. al, 2021
>
>  [2] Cumulative cultural learning: Development and diversity, Cristine H. Legare, 2017

---

> > ### Comment · Reviewer_9itL · 2024-08-11
> >
> > Dear Authors,
> >
> > Thank you for your detailed and thoughtful response to my initial comments. I appreciate the clarifications you have provided regarding the complexity, scalability, and potential real-world applications of your models. Your explanations have helped to address many of the concerns I raised, and I commend you for the thoroughness of your rebuttal.
> >
> > I would like to offer a few additional comments and suggestions:
> >
> > Complexity and Computational Efficiency:
> >
> > Your clarification on the minimal changes required to existing RL algorithms and the efficiency gains from cultural evolution is valuable. It might be beneficial to highlight these aspects more prominently in the manuscript to preempt concerns about complexity and resource demands from other readers as well. This could be particularly useful in the introduction or discussion sections to frame the work within the context of practical RL implementations. Please ensure that these details are updated in the paper.
> >
> >
> > Scalability and Broader Applications:
> >
> > The results you presented for Atari Breakout, Freeway, and Brax Half-Cheetah are promising. Including these results in the main text, or at least in an appendix, could strengthen the paper by providing concrete evidence of the models' scalability beyond the initial set of environments. Additionally, a more detailed discussion on how these findings might generalize to other real-world tasks would enhance the paper’s impact. Please consider updating the manuscript to reflect these new results and discussions.
> > Real-World Applications:
> >
> > Your response about modeling human cultural accumulation “in-silico” is compelling. To further enrich this discussion, you might consider elaborating on how these models could be adapted or extended to address specific real-world challenges. For example, could the approach be tailored to specific industries or applications, such as healthcare or autonomous systems? Including a few concrete examples or case studies could make the potential impact of your work even clearer. It would be valuable to update the paper with these considerations.
> >
> >
> > Balancing Social Learning and Independent Discovery:
> >
> > The insights you provided on balancing social learning and independent discovery, particularly regarding the role of oracle noise, are intriguing. It could be useful to explore this balance further in your experiments or discussion, perhaps by analyzing the sensitivity of your models to different levels of oracle noise or by suggesting guidelines for selecting this parameter in practice. Incorporating these insights into the paper would provide further clarity for the readers.
> >
> >
> > Overall, I believe your paper makes a significant contribution to the field of reinforcement learning, and these additional points are intended to further strengthen its presentation and impact. I recommend that the paper be updated with these additional details to ensure that it fully reflects the comprehensive nature of your work.
> >
> > I have decided to leave my score unchanged.
> >
> >
> > Thank you

---

> > > ### Author Response · Authors · 2024-08-12
> > >
> > > Thank you for your kind response and for highlighting points worth clarifying in the updated manuscript; we greatly appreciate the time taken to engage with our work in depth and provide suggestions that will no doubt improve the quality of the final version.
> > >
> > > We are glad that you see our paper as providing a significant contribution and thank you again for your efforts in ensuring its impact by strengthening its presentation.

---

### Official Review · Reviewer_2ZBi · 2024-07-11

**Soundness:** 2
**Presentation:** 2
**Contribution:** 3
**Rating:** 5
**Confidence:** 3

**Summary:**

The paper studies cultural accumulation within the context of RL agents. The techniques involve social learning based on in-context learning and in-weights learning. Each agent is modelled as a POMDP within a larger POSG. The techniques are applied to memory sequence, goal sequence, and TSPs, where they are compared to RL^2.

**Strengths:**

The concept of transmitting information across generations other than the genotype is underexplored, which the authors study using in-context accumulation (i.e. passing on hiddent states).

The results show a positive performance trend.

**Weaknesses:**

It does not appear obvious how to implement such a training setup on a physical system.

Traditional evolutionary algorithms pass policy parameters from the previous generation to the next, and these techniques are not discussed. And indeed, there is a big field integrating RL techniques with evolutionary algorithms.

Observing other agents is also not new and this is commonly seen in multi-agent reinforcement learning. For instance, sharing observations:
https://arxiv.org/pdf/1812.00922
The benefit of sharing hidden states is not clear.

TSPs are best solved by evolutionary algorithms. There is no comparison to such algorithms.

My initial reading of the POSG section was that there is an error in the definition of the reward function as it does not index the individual. However, as mentioned in the Limitations section, the authors basically use the same R for all agents. So I believe that indexing R_i and then mentioning that all the R_i’s are the same would be more clear. It does cast some doubt on why the POSG formalism is more suitable.

It is said that each member of a given generation learns on an independent instance of the environment with frozen agents from the previous generation and that for each agent inidividually, the problem is a POMDP. So there appears to be no benefit in introducing POSGs.

Baselines: only one baseline RL^2. And the performance difference can likely be explained by the assumption of a full state oracle. It does not seem a fair comparison to assess the algorithms with different environment access assumptions.
“To facilitate (1) during training, but not at test time, we assume access to an oracle −i with fixed −i parameters θ̂ . This oracle is able to condition on the full state s, which we refer to as privileged information. Agent i can observe the behaviours of −i as they jointly act in an environment.”

The RL^2 seems to be an essential part of the algorithm (Appendix A) yet it is not explained. Moreover, RL^2 too passes on hidden state information so the comparative benefit should be explained as well.

The need for two separate phases of evolution is not clearly explained. Also it is not clear how they connect.

It is not clear how RL^2 and the proposed technique are comparable in terms of the total number of evaluations but also in the observations and learning setting.
Algorithm 1:
 p_obs, phi, and theta are not being used

 there is no need to repeat line 19 for all i. It can be put at the end of the generation since it is the same for all i.

 l.11: index n is undefined

There is limited theory development.

There is no explanation of why the noisy oracle can work well.

**Questions:**

The authors mention the two variants of RL^2 show the learning for the length of a single lifetime as well as the full length combining generations (equivalent to the length of their own algorithm). If so, then how come Figure 4 shows the same length for RL^2 in both cases?

Why does the noisy oracle work well? Is this because in the evaluation phase, only an observation can be provided so there is less discrepancy between the phases?

“We model cultural accumulation as taking place within POSGs. Each member of a given generation gm+1 learns in parallel on an independent instance of the environment, which also contains static members of the previous generation, gm.  Therefore, from the perspective of a given agent i in gm+1 this is a POMDP, since the prior, static generation can be considered part of the environment.”
Can this be clarified? For example, does this mean the policies of other agents are frozen? And why do we need the framework of POSGs if actually the problem to solve is only POMDP.

What is the importance of having the POMDPs being sampled from a distribution to demonstrating the proposed approach? I don’t clearly get this from the text. Also, it appears that actually also the environment is just a POMDP with observations indicating the task.

What are the hyperparameters for RL^2

What is the benefit of sharing the recurrent state at one step vs sharing the observation at each time step?

How would you implement this in a physical setup?

**Limitations:**

The authors mention limitations but not the strongest ones:
    • the presence of the full-state oracle.
    • The applicability

---

> ### Author Rebuttal · Authors · 2024-08-06
>
> We would like to thank the reviewer for their thorough and extensive review. We are pleased that the reviewer finds that “the concept of transmitting information across generations other than the genotype is underexplored” and that “the results show a positive performance trend.”
>
> # On related work
>
> > Traditional evolutionary algorithms...
>
> The core distinction between *cultural evolution* and traditional evolutionary algorithms (i.e., genetic algorithms) is that cultural evolutionary algorithms [1] accumulate knowledge and skills, as opposed to parameters or DNA, that are relevant to the population and explore directly in this space of this knowledge and skills. We additionally explore passing on policy parameters in Figure 5 (right).
>
> > Observing other agents is also not new...
>
> We acknowledge that observing other agents is not new. As seen in the paper in Table 1, we very deliberately highlight where our work sits in relation to the literature, as acknowledged by reviewers h1KJ and R6qB. There we draw attention to the fact that our work uniquely combines implicit, third person imitation with generational training to achieve cultural accumulation.
>
> # On the formalism
>
> > My initial reading of the POSG section was that there is an error in the definition of the reward function.
>
> Thanks for catching this! We’ve fixed it in our manuscript.
>
> > The problem is a POMDP, so there appears to be no benefit in introducing POSGs.
>
> We introduce POSGs to introduce the notation we will be using to refer to other agents and their policies. Indeed, our setting is a POMDP from the perspective of any individual agent. However, when describing the implementation and method with formal notation, it is far easier to use the notation from POSGs.
>
> # On baselines
>
> > Only one baseline RL^2. And the performance difference can likely be explained by the assumption of a full state oracle.
>
> As stated in Section 3 of the paper, the goal was to compare to single-lifetime baselines using an otherwise equivalent algorithm, as the method of accumulation is additive (i.e., builds atop the same underlying RL algorithm). As RL^2 is the meta-RL algorithm underlying our in-context accumulation implementation, that same algorithm without accumulation is the natural baseline for showing the impact of test time accumulation. The full-state oracle is only for training in the in-context regime. Training with an oracle for the RL^2 baselines actually impedes performance because we do not assume access to oracles at test time.
>
> > The RL^2 seems to be an essential part of the algorithm (Appendix A)...
>
> In the background section on Meta-RL (Section 2.3), we introduce the concepts underpinning RL^2 and cite the paper appropriately, but do not explicitly introduce it as RL^2 there, which we now do in our updated manuscript.  Appendix A provides the algorithm for RL^2 in detailed pseudocode, with our adaptations to enable in-context accumulation in red.
>
> # Points of clarification
>
> > The need for two separate phases of evolution is not clearly explained...
>
> Could the reviewer please clarify what they mean by this? If referring to the difference between in-context and in-weights accumulation, the former involves accumulation over in-context learning agents, whilst the latter involves accumulation over agents that are training and is most similar to prior work on generational training [3].
>
> > It is not clear how RL^2 and the proposed technique are comparable in terms of the total number of evaluations...
>
> We apologise for the insufficient clarity here. As Algorithm 2 is referenced first, but appears in the Appendix, we used the line “rollout agents as in training” to refer to the lines in Algorithm 2 that show how p_obs, phi and theta are used. We have rectified this by including those lines of pseudocode explicitly in Algorithm 1 as well. The total number of evaluations is the same for RL^2 and in-context accumulation. We have updated our manuscript to make this more explicit.
>
> > There is no explanation of why the noisy oracle can work well.
>
> The noisy oracle has full state information. We add some noise to this state information so that the oracle is sub-optimal. Its performance is plotted simply to illustrate that in-context accumulation at test time goes beyond the performance of anything seen in training, which provides further evidence that in-context accumulation outperforming RL^2 baselines is not simply “explained by the full-state oracle”.
>
> # Addressing questions:
>
> 1. The two RL^2 baselines have different contexts (i.e., number of trials within an episode) during training. For fairness, we evaluate over the same total length in case the shorter-context baseline generalises zero-shot to longer contexts, which we do not observe happen.
>
> 2. See Points of clarification.
>
> 3. See On clarity.
>
> 4. Sampling from a space of POMDPs corresponds to randomly sampling a specific environment instance.
>
> 5. They are the same as for the corresponding accumulation algorithm. This is because the underlying RL algorithm is the same between the baselines and accumulation. We have updated our manuscript to explicitly state this.
>
> 6. The recurrent state is not shared. It is used to distinguish between generations for in-context accumulation and plays no role in in-weights accumulation. Sharing observations is an instance of explicit communication/ information transmission, whereas we consider implicit third person social learning for its flexibility in allowing agents to learn from the behaviour of other agents, or independently explore or exploit.
>
> 7. An example would be teams of agents at deployment time figuring out how to improve on solutions to tasks e.g. in a warehouse order fulfilment setting.
>
> [1] A comprehensive survey on cultural algorithms, Alireza Maheria, et. al, 2011
>
> [2] A Survey of meta-reinforcement learning, Jacob Beck, et. al, 2023
>
> [3] Open-ended learning leads to generally capable agents, Stooke, et. al, 2021

---

> ### Comment · Reviewer_2ZBi · 2024-08-09
>
> Thanks for the response. I have some further comments.
>
> I think it should be made more clear in the text that the figures relate to the evaluation and not the training phase. Also, if there is no distinction between the evaluation and the training phase, how come even in in-weights training, the plot for the single-lifetime training is essentially a fixed line even though the agents have the same number of environment interactions? We currently have no view on the learning curves of RL^2.
>
> There is still only one comparison in the paper, to RL^2, which in a way is an ablation. An ablation study is important but this is quite minimal.  There are potentially a lot multi-agent RL techniques to compare to but this has not been done. Empirical comparisons to techniques for POSGs, implicit and/or explicit communication in multi-agent RL, or indeed cultural transmission across generations, would be interesting to see but are absent. Also, it is not clear what are the comparative scores of the techniques compared to the state-of-the-art.
>
> Last, the authors show the effect of different noise levels but do not seem to mention which noise level is used for the results in the evaluation plots.
>
> While I appreciate the authors’ extensive responses and additional experiments, the claims remain overly broad and it is not clear how well the method fairs when compared to algorithms making similar assumptions.

---

> ### Author Response · Authors · 2024-08-09
> **Taking on feedback and responding to comments**
>
> Thank you for taking the time to read our rebuttal and for sharing your further concerns. We strongly believe that these are addressable and endeavour to do so within this comment.
>
> > It should be made more clear in the text that the figures relate to the evaluation and not the training phase.
>
> We take this onboard and adapt our captions accordingly.
>
> > How come even in in-weights training, the plot for the single-lifetime training is essentially a fixed line?
>
> Thank you for pointing this out! To ensure that our single-lifetime baseline is fair, we use the same population size (i.e., number of seeds) as for accumulation and report the best performing agent's (i.e., argmax over seeds) maximum achieved return over the duration of its training as the dashed single-lifetime baseline. We now explicitly state this in the manuscript.
>
> > We currently have no view on the learning curves of RL^2.
>
> We can provide these in the Appendix. We would like to have been able to include them in the rebuttal if they are of importance to the reviewer, but they were not initially requested and we cannot upload new figures or amend the pdf during the discussion period.
>
> > There are potentially a lot multi-agent RL techniques to compare to [...] techniques for POSGs, implicit and/or explicit communication in multi-agent RL...
>
> We would like to humbly remind the reviewer that we use the setting of a POSG to introduce the formalism for our algorithms before explaining how the setting reduces to a POMDP because the policies of previous generations are fixed. This means that we are training successive single-agent RL policies with PPO and using the methods we introduce to achieve performance improvements via cultural accumulation. For in-weights accumulation, the baseline of partial network resets [1] therefore makes sense, which we show *additionally* improves accumulation itself in Figure 5 (right).
>
> Approaches to implicit communication in MARL [2, 3] assume that agents are incentivised to *learn to communicate* due to cooperative rewards. We make no such assumptions, as each agent in our study is simply maximising its own independent reward. We believe that combining our work with implicit or explicit learned communication in cooperative settings would be an exciting direction for future work.
>
> > ...or indeed cultural transmission across generations...
>
> Could the reviewer please clarify what they mean by a baseline here? We are precisely exploring cultural transmission across generations, where previous work has explored a single step (i.e., one generation) of cultural transmission [4, 5].
>
> > It is not clear what are the comparative scores of the techniques compared to the state-of-the-art.
>
> We would like to emphasise that prior work on social learning and cultural transmission in RL [4, 5] has focused on learning from an expert for a single generation of transmission, both in a setting similar to our in-weights setting [4] and where human experts provide demonstrations at test time [5]. In both cases, the reported baselines are ablations of their method, as the purpose was to demonstrate that social learning can be achieved and provide performance benefits in RL. In our work, we extend these ideas to span multiple generations, whilst making no additional assumptions and removing the assumption of having an expert at test time from [5] by creating this generational bootstrap.
>
> The goal is therefore not to achieve state-of-the-art performance on a set of benchmarks, but to demonstrate that cultural accumulation can be modelled in RL and that it can reap performance benefits relative to running the same algorithm for one continuous lifetime.
>
> In an effort to further address this concern, we provide the final performance of successive generations of policy distillation (i.e., generational training) [6] in comparison to the final performance at each generation of in-weights accumulation.
>
> | Task                           | Generational Training     | In-Weights Accumulation |
> |-------------------------|-----------------------------|------------------------------|
> | Memory Sequence    | 0.6, 2.7, 6.6, 7.5, 8.5      | 0.6, 4.2, 7.9, 8.3, 11.4      |
> | Goal Sequence         | 0.7, 1.8, 2.9, 3.1              | 0.7, 3.7, 4.2, 4.3               |
>
> > ...do not seem to mention which noise level is used for the results in the evaluation plots.
>
> We take this onboard and now include them in our stated hyperparameters.
>
> [1] The primacy bias in deep reinforcement learning, Nikishin et. al, 2022
>
> [2] Learning to communicate implicitly by actions, Tian et. al, 2020
>
> [3] Foraging via multi-agent RL with implicit communication, Shaw et. al, 2021
>
> [4] Emergent social learning via multi-agent reinforcement learning, Ndousse et. al, 2021
>
> [5] Learning few-shot imitation as cultural transmission, Bhoopchand et. al, 2023
>
> [6] Open-ended learning leads to generally capable agents, Stooke et. al, 2021

---

> ### Author Response · Authors · 2024-08-11
>
> We would like to kindly ask that the reviewer considers our most recent comment and ask if they have any further questions or concerns.

---

> > ### Comment · Reviewer_2ZBi · 2024-08-12
> >
> > I commend the authors for their thorough discussion and additional work during this period. There are no major red flags so I will increase my score to 5.

---

> > > ### Author Response · Authors · 2024-08-12
> > >
> > > We thank the reviewer for engaging in this discussion, considering our responses and raising their score. We are certain that the points raised on clarity and the additional baseline run in response to the reviewer's comments will have a positive impact on the quality of the final manuscript.

---

### Author Rebuttal · Authors · 2024-08-06

We are grateful to the reviewers for their insightful feedback. We appreciate the consensus that our work is exploring an important, understudied area and that our results indicate positive progress by demonstrating that cultural accumulation can outperform single-lifetime baselines. This is the key takeaway of our work, and we hope it will accelerate future research on cultural accumulation in learning agents.

In particular, we are glad that reviewers found that “the approach is novel and interesting” (R6qB), “the work is well situated among related prior work” (R6qB), “the experiments are well-designed to test the hypotheses, with clear evidence showing the benefits of cultural accumulation” (9itL), and “the results are significant” (h1KJ), showing “a positive performance trend” (2ZBi).

Some reviewers raised concerns with clarity (h1KJ, R6qB) and there were a few misinterpretations of parts of the paper, further indicating that the presentation of the algorithms could be clearer. We strongly believe that this is addressable and would like to emphasise that reviewers have not raised any further common issues with our work.

# On clarity

We greatly appreciate the reviewers’ perspectives on how the paper, which “provides a thorough analysis of both in-context and in-weights accumulation, exploring different environments and scenarios” (9iTL), can be made clearer in its presentation of the different algorithms and results. In particular, we agree that more algorithmic details (many of which currently feature in pseudocode within the appendix) should be included within the main text. We will ensure to do so for a camera-ready version of the paper, in favour of moving some results to the Appendix and/or slightly shortening the Background section if necessary.

We are confident that this will strengthen the manuscript’s overall quality and hope that it adequately addresses this common concern raised by some of the reviewers.

For new results corresponding to specific reviewer's comments, please see the attached pdf document.

---

### Decision · Program_Chairs · 2024-09-25

**Decision:**

Accept (poster)

**Comment:**

This paper investigates the hypothesis that cultural accumulation may be possible in RL. In other words, agents may benefit not only from their own experiences while interacting with a given environment but also from knowledge implicitly passed down from previous generations. The authors proposed, investigated, and empirically analyzed two ways in which accumulation could occur.

All reviewers had an overall positive impression of this work, highlighting the well-conducted experiments designed to test the hypotheses investigated by the authors (as mentioned, e.g., by 9itL, 2ZBi, and h1KJ). Reviewer R6qB and others also argued that the proposed approach is novel and interesting. The reviews were thorough and insightful, and the discussion phase helped clarify various important concerns raised by the reviewers—particularly claims made by the authors that were perceived as overly broad and unclear. Some reviewers, such as 2ZBi, also argued (even post-rebuttal) that it is not immediately apparent how well the proposed approach compared to other methods that made similar assumptions.

Overall, however, the consensus among the reviewers was that this paper does contribute important insights that may benefit the NeurIPS community and that points of contention that were brought up were not critical and are addressable in a revised version of the paper. The reviewers strongly encouraged the authors to improve their document by adding further discussion and improving clarity to ensure that the insights introduced in this work become more apparent to a broader audience.